# Cytosolic ROS production by NADPH oxidase 2 regulates muscle glucose uptake during exercise

Carlos Henríquez-Olguin [1,2], Jonas R. Knudsen [1], Steffen H. Raun[1], Zhencheng Li[1], Emilie Dalbram [3], Jonas T. Treebak [3], Lykke Sylow[1], Rikard Holmdahl [4], Erik A. Richter [1], Enrique Jaimovich[2] & Thomas E. Jensen [1]*

Reactive oxygen species (ROS) act as intracellular compartmentalized second messengers, mediating metabolic stress-adaptation. In skeletal muscle fibers, ROS have been suggested to stimulate glucose transporter 4 (GLUT4)-dependent glucose transport during artificially evoked contraction ex vivo, but whether myocellular ROS production is stimulated by in vivo exercise to control metabolism is unclear. Here, we combined exercise in humans and mice with fluorescent dyes, genetically-encoded biosensors, and NADPH oxidase 2 (NOX2) loss-of-function models to demonstrate that NOX2 is the main source of cytosolic ROS during moderate-intensity exercise in skeletal muscle. Furthermore, two NOX2 loss-of-function mouse models lacking either p47phox or Rac1 presented striking phenotypic similarities, including greatly reduced exercise-stimulated glucose uptake and GLUT4 translocation. These findings indicate that NOX2 is a major myocellular ROS source, regulating glucose transport capacity during moderate-intensity exercise.

---

[1] Department of Nutrition, Exercise and Sports, Section of Molecular Physiology, University of Copenhagen, Universitetsparken 13, 2100 Copenhagen, Denmark. [2] Center for Exercise, Metabolism and Cancer, ICBM, Universidad de Chile, 8380453 Santiago, Chile. [3] Novo Nordisk Foundation Center for Basic Metabolic Research, Integrative Metabolism and Environmental Influence, Faculty of Health and Medical Sciences, University of Copenhagen, Blegdamsvej 3A, 2200 Copenhagen, Denmark. [4] Section for Medical Inflammation Research, Department of Medical Biochemistry and Biophysics, Karolinska Institute, Solnavägen 9, 171 65 Solna, Sweden. *email: TEJensen@NEXS.ku.dk

A single bout of exercise prompts a rapid adaptive increase in energy substrate metabolism in contracting skeletal muscle to meet the increased demand for ATP production. Understanding the molecular signal transduction pathways that orchestrate these metabolic responses in muscle has broad implications, from optimizing athletic performance, to understanding fundamental stress-adaptive responses at the cellular level, to prevention and treatment of aging-related and lifestyle-related diseases in muscle and other tissues[1].

The production of oxygen-derived free radicals and derivatives thereof, collectively referred to as reactive oxygen species (ROS), increases in skeletal muscle during exercise and has long been considered to mediate adaptive responses to both acute bouts of exercise and chronic exercise training[2]. Although historically viewed as a by-product of oxidative metabolism in mitochondria, it has been suggested that ROS may be produced enzymatically by extra-mitochondrial sources in contracting muscle, including NADPH oxidase (NOX)[3] and xanthine oxidase[4]. However, the primary ROS source in the context of physiological in vivo exercise remains uncertain due to the difficulty in measuring and quantifying exercise-stimulated ROS production[5].

Research in the past 60 years has established that glucose transport in vivo is controlled by a coordinated increase in glucose delivery, glucose transport into muscle fibers and intracellular metabolism[6,7]. Among these events, the insulin-independent translocation of glucose transporter protein 4 (GLUT4) from intracellular storage depots to the cell surface to facilitate glucose entry appears to be a key molecular event[8,9]. Studies in isolated ex vivo incubated skeletal muscles from rodents have shown that glucose transport in response to electrically stimulated contraction and mechanical stress is lowered by antioxidants[10,11], suggesting that ROS are required for increased glucose transport[12]. Interestingly, the small GTPase Rac1 is activated by muscle contraction and passive stretch, and is necessary for both stimuli to increase glucose uptake in isolated mouse muscles[13,14]. Rac1 is best known to bind and orchestrate regulators of actin remodeling[15], but the recruitment of GTP-loaded active Rac GTPase isoforms in conjunction with a complex of other NOX2 regulatory proteins (p67phox, p47phox, and p40phox) is also essential to stimulate superoxide ($O_2^-$)-production by the membrane-bound NOX2 complex[16]. However, whether NOX2-induced ROS production regulates muscle glucose uptake in vivo and if the reduced exercise-stimulated glucose uptake observed in Rac1-deficient muscles[17] is due to reduced ROS production, is currently unknown.

In the present study, we took advantage of recent methodological developments in the redox signaling field, including a method for preservation of in vivo ROS modifications and genetically encoded ROS biosensors[18]. This allowed us to measure both general and localized NOX2-specific ROS production in wild-type (WT) mice and mice lacking NOX2 activity due to the absence of either the Rac1 or p47phox regulatory subunits. Using these approaches, we show that NOX2 is activated by moderate-intensity exercise and is the predominant source of ROS production under these conditions. Moreover, a large reduction in exercise-stimulated glucose uptake and GLUT4 translocation were shared features between the two NOX2 loss-of-function models. Collectively, these results imply that NOX2 is a major source of ROS production during moderate-intensity exercise and that Rac1 is required for GLUT4 translocation and glucose uptake due to its essential role in NOX2 activation.

## Results

### Moderate-intensity exercise causes a pro-oxidative shift in human muscle. Plasma redox markers are increased during both moderate-intensity and high-intensity exercise in humans[19]. However, whether muscle ROS production increases during moderate-intensity exercise in humans is unknown. To explore this, we estimated total oxidant production in skeletal muscle from healthy young men (Supplementary Table 2) before and after a single 30 min bout of bicycle ergometer exercise at a moderate 65% peak power output exercise intensity using the redox-sensitive dye dichlorodihydrofluorescein diacetate (DCFH). We found that exercise stimulated an 86% increase in DCFH oxidation in human skeletal muscle (Fig. 1a), which was accompanied by an expected increase in the phosphorylation state of known exercise-responsive proteins (Fig. 1b). No changes in total protein expression were observed (Supplementary Fig. 1A). This shows that moderate-intensity exercise is pro-oxidative in human skeletal muscle.

### The exercise-induced pro-oxidative shift measured by DCFH requires NOX2 activity in mice. To dissect the ROS source during exercise, we used a previously described p47phox-mutated mice (ncf1*) harboring a loss-of-function mutation in the regulatory NOX2 subunit, p47phox[20]. An acute moderate-intensity treadmill exercise bout (65% maximal running speed) for 20 min increased DCFH oxidation (+45%) in WT mice, a response that was completely abolished in p47phox-deficient ncf1* mice (Fig. 1c). This observation was found to be independent of alterations in antioxidant enzyme abundance in TA muscles from ncf1* mice compared to WT mice (Supplementary Fig. 1B). This shows that NOX2-activity is required for moderate-intensity exercise-induced DCFH oxidation in mice.

### Exercise-stimulated p38 MAPK$^{Thr180/Tyr182}$ phosphorylation is reduced in ncf1* quadriceps muscle. Exercise-induced ROS production has been suggested to activate a number of kinases linked to glucose uptake-regulation in skeletal muscle[21]. Interestingly, p-p38 MAPK$^{Thr180/Tyr182}$ levels were lower in quadriceps muscle of ncf1* mice compared to WT mice after exercise (Fig. 1d), but not in TA (Supplementary Fig. 1C) or soleus muscles (Supplementary Fig. 1D). The phosphorylation of ERK$^{Thr202/Tyr204}$, AMPK$^{Thr172}$, and its substrate ACC$^{Ser212}$ did not differ significantly between genotypes (Fig. 1d and Supplementary Fig. 1C, D). We also measured SERCA, eEF2$^{Thr57}$, CaMKII$^{Thr287}$, and TBC1D1$^{Ser231}$ but found no genotype-difference (data not shown). No genotype-difference was observed for total AMPKα2, p38 MAPK, ERK 1/2, or ACC in any of the studied muscles (Fig. 1e and Supplementary Fig. 2). The variable responsiveness of these kinases supports that the mice performed moderate-intensity exercise and shows that NOX2 is not consistently required for activation of these kinases.

### NOX2 is required for increased cytosolic ROS during exercise. Since DCFH oxidation does not provide information about the ROS source, we further investigated the subcellular redox changes in response to exercise in skeletal muscle using a recently described redox histology method[18] (see graphical depiction in Supplementary Fig. 3A). This method enables the preservation and visualization of the redox state of a transfected redox-sensitive GFP 2 (roGFP2)-Orp1, targeted to cytosolic and mitochondrial compartments following in vivo exercise. The Orp1 domain facilitates roGFP2 oxidation in the presence of $H_2O_2$ to elicit a ratiometric change, with an increase in 405 nm and a decrease in 470 nm fluorescence. As such, the ratio between the two wavelengths is a measure of $H_2O_2$ production in the targeted compartment, here the cytosol and the mitochondria[22].

Mito-roGFP2-Orp1 was located in the mitochondrial compartment (co-localizing with tetramethylrhodamine, ethyl ester,

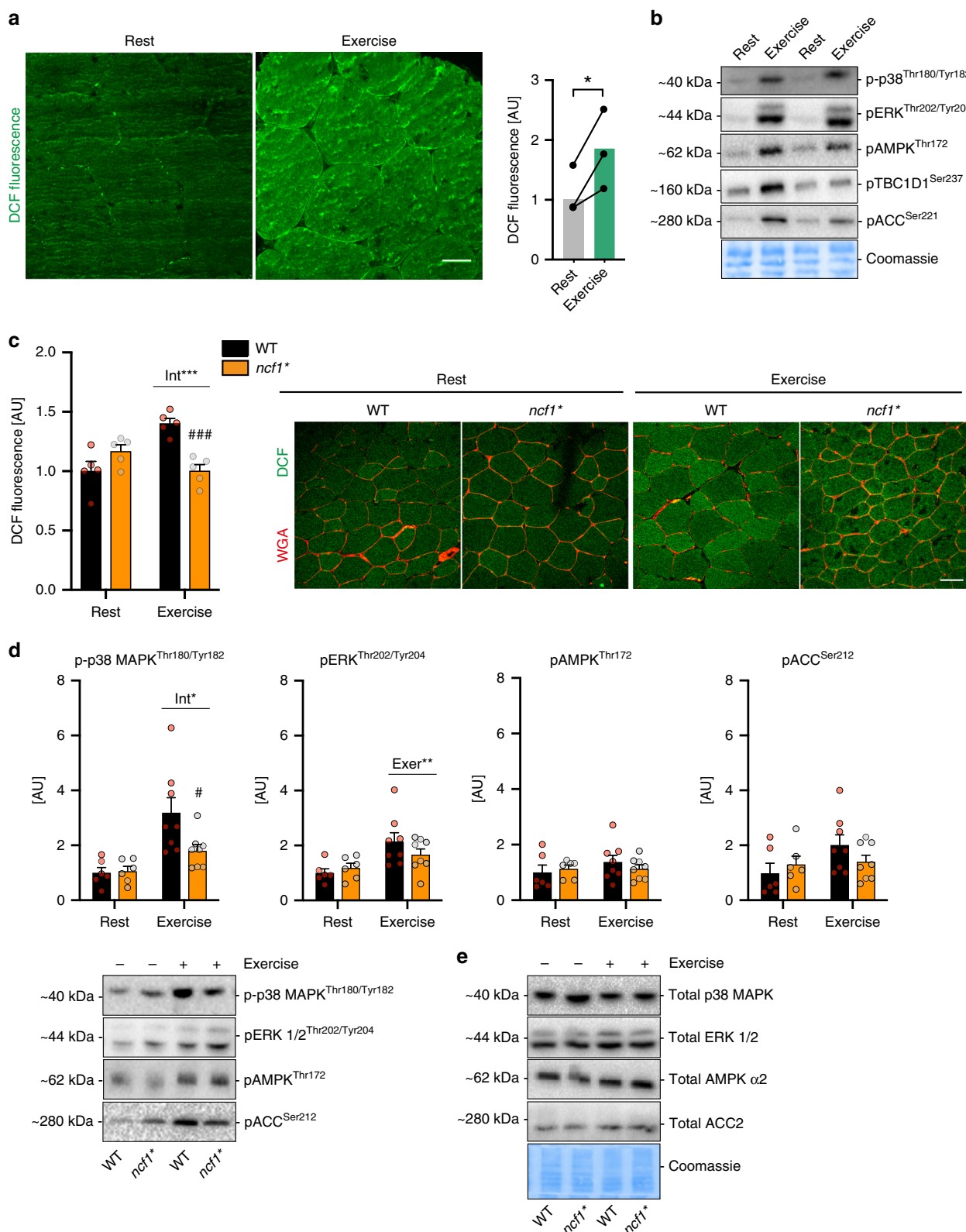

Supplementary Fig. 3B, C) and shown to be sensitive to $H_2O_2$ (Supplementary Fig. 3D). Interestingly, oxidation of Mito-roGFP2-Orp1 probe was lowered similarly in both genotypes by exercise (Fig. 2a). In contrast, cytosolic roGFP2-Orp1 oxidation showed a main effect of genotype (Fig. 2b), driven by an exercise-induced increase in roGFP2 oxidation in WT but not $ncf1^*$ mice.

To substantiate that NOX2 activity was required for exercise-stimulated ROS production, we electroporated a biosensor designed to measure NOX2 activity, the p47roGFP construct, into muscles from inducible muscle-specific Rac1 knockout mice (Rac1 imKO), which are predicted to lack functional NOX2 complex. Treadmill exercise caused an acute increase in p47roGFP oxidation in WT TA muscle which was completely

**Fig. 1** Moderate-intensity exercise causes a pro-oxidative shift in human and murine muscle. **a** 2′,7′-dichlorodihydrofluorescein diacetate (DCFH) oxidation (DCF, $n = 3$)) and **b** exercise signaling in human vastus lateralis before and after a bout of moderate-intensity cycling (30 min, 65% maximal power output) in young-healthy volunteers ($n = 3$). **c** Representative images and quantification of the exercise-stimulated (20 min, 65% maximal running speed) DCFH oxidation in tibialis anterior muscle from WT and *ncf1\** mice ($n = 5$ per group). **d** Exercise signaling in quadriceps muscle from WT and *ncf1\** mice ($n = 6$ resting and $n = 8$ exercising group). **e** Total proteins levels of p38 MAPK, ERK 1/2, Total alpha2 AMPK, Total ACC, and Coomassie staining as loading control. For **a** paired *t*-test was performed for statistical analysis, * denotes $p < 0.05$ compared to resting condition. For **c, d** a two-way ANOVA was performed to test for effects of exercise (Exer) genotype (Geno), and interaction (Int), followed by Tukey's post hoc test with correction for multiple comparisons. *, **, *** Denotes $p < 0.05$, $p < 0.01$, and $p < 0.001$, respectively, for main effects/interactions. #, ### Denotes $p < 0.05$ and $p < 0.001$ compared to the WT group. Individual values and mean ± standard error of the mean (SEM) are shown. Scale Bar = 50 μm. For **b** western blots for total proteins are shown in Supplementary Fig. 1, uncropped blot and quantifications are shown in the Source data file. For **c–e**, source data are provided in the Source Data file

absent in Rac1 KO mice, showing that Rac1 is essential for NOX2 activation in skeletal muscle (Fig. 2c). A similar dependence of NOX2 activation on Rac1 was observed in electrically stimulated FDB fibers in vitro (Supplementary Fig. 3H). The absence of p47roGFP oxidation in Rac1 imKO muscles was not explicable by differences in antioxidant enzyme abundance (Supplementary Fig. 3I). In accordance, exercise-stimulated DCFH oxidation was completely absent in Rac1 imKO compared to WT littermates (Fig. S2G) with no differences under resting conditions (Supplementary Fig. 2I).

Collectively, these results show that NOX2 is activated and constitutes a major source of cytosolic ROS production during endurance-type exercise in skeletal muscle. In contrast, mitochondrial ROS production is lowered during acute exercise independently of NOX2.

**NOX2 is required for exercise-stimulated glucose uptake**. Given that Rac1 imKO mice display a severe reduction in treadmill exercise-stimulated glucose uptake and GLUT4 translocation[17], we reasoned that if Rac1 was acting via NOX2 then *ncf1\** mice should show a similar reduction. We first conducted a general characterization of *ncf1\** mice. Similar total body weight but reduced fat mass was observed in *ncf1\** mice compared to age-matched WT mice (Supplementary Fig. 4). Respiratory exchange ratio (RER) was lower in *ncf1\** mice during the light phase compared to WT mice (Fig. 3a), despite a similar habitual activity observed between genotypes (Fig. 3b). Thus, *ncf1\** mice, similar to Rac1 imKO mice[23], display a shift towards fat oxidation at rest and during fasting.

Next, we determined whether NOX2-dependent ROS generation is required for exercise-stimulated glucose uptake. Each mouse was exercised at a relative work load corresponding to 65% of its maximum running capacity for 20 min Exercise-induced 2-[³H] deoxyglucose (2DG) uptake was markedly attenuated in quadriceps, soleus, and TA muscles from *ncf1\** mice compared to WT (Fig. 3c–e). Importantly, the reduction of exercise-stimulated glucose uptake in *ncf1\** mice was not due to differences in maximal running capacity (Fig. 3f), blood glucose levels (Fig. 3g), or intramuscular energetics evaluated as $NAD^+$ and NADH levels (Fig. 3h, i). However, plasma lactate levels were increased by exercise in WT but not in *ncf1\** mice (Fig. 3j), consistent with a reduced reliance on glycolysis for energy production. No differences were observed in muscle fiber size (Fig. 4a–c), fiber-type composition (Fig. 4d, e), mitochondrial complex expression (Fig. 4f, g, quantifications in Supplementary Fig. 5) or capillary density (Fig. 4h). Thus, *ncf1\** in comparison to WT mice were markedly impaired in their ability to stimulate glucose uptake in vivo across multiple muscles, without indications of altered glucose delivery or oxidation capacity.

**NOX2 is required for exercise-stimulated GLUT4 translocation**. Next, we tested if the reduction in exercise-stimulated glucose uptake in *ncf1\** mice could be explained by reduced GLUT4 translocation as observed in the Rac1 imKO[17]. Consistent with this notion, the exercise-stimulated increase in surface-membrane GLUT4-GFP-*myc* in TA muscle from WT was virtually abolished in *ncf1\** mice (Fig. 5a). Importantly, total protein abundance of GLUT4 and HKII did not differ between genotypes (Fig. 5b–d, quantification in Supplementary Fig. S6). Taken together, these data demonstrate that *ncf1\** mice phenocopy the Rac1 imKO mice in terms of pronounced impairments in exercise-stimulated muscle glucose uptake and GLUT4 translocation[17]. This strongly suggests that the common denominator between p47phox and Rac1, NOX2 activation, is required for exercise-stimulated glucose uptake and GLUT4 translocation.

## Discussion

The source of ROS during exercise and the role of ROS in the acute and chronic adaptations to exercise, in particular in the context of non-damaging moderate-intensity in vivo exercise, has remained undetermined for more than 30 years[24]. The current study showed that moderate-intensity endurance exercise acutely caused a pro-oxidative shift in both humans and mice. Strikingly, our results demonstrated that NOX2 was required for this shift and was a major cytosolic ROS source during moderate-intensity exercise. Furthermore, NOX2 activity was necessary for exercise-stimulated GLUT4 translocation and glucose uptake in skeletal muscle, providing a novel molecular explanation of why Rac1 imKO mice have a strong impairment of exercise-stimulated GLUT4 translocation and glucose uptake.

The myocellular ROS source(s) during exercise conditions has long been debated, with both cytosolic and mitochondrial sources having been proposed based on pharmacological in vitro/ex vivo studies[3,25–28]. Presently, we observed an increase in oxidants measured by DCFH oxidation in exercising mouse and human muscle. Remarkably, despite the non-specific nature of DCF fluorescence, we found that the increase in exercise-stimulated DCFH oxidation was virtually abolished in mice lacking either Rac1 or p47phox components of NOX2. It should be noted, though, that there might be a potential species-difference in DCFH staining pattern, with the human staining for unknown reasons seeming less homogeneous than the mouse staining. To further investigate the contribution of different intramyocellular ROS sources during exercise in mice, we used genetically encoded redox probes targeted to different myocellular compartments to study the ROS sources during in vivo exercise. Exercise promoted an oxidation of cytosol-targeted roGFP2-Orp1 biosensor in WT but not in NOX2-deficient mice. On the other hand, mitochondria-targeted roGFP2-Orp1 probe oxidation was reduced by exercise independently of NOX2. The NOX2 complex-targeted p47roGFP biosensor oxidation was increased by exercise in WT muscles but not in Rac1-deficient mice. Overall, this demonstrates that a functional NOX2 complex, but not mitochondria, is required for cytosolic ROS production

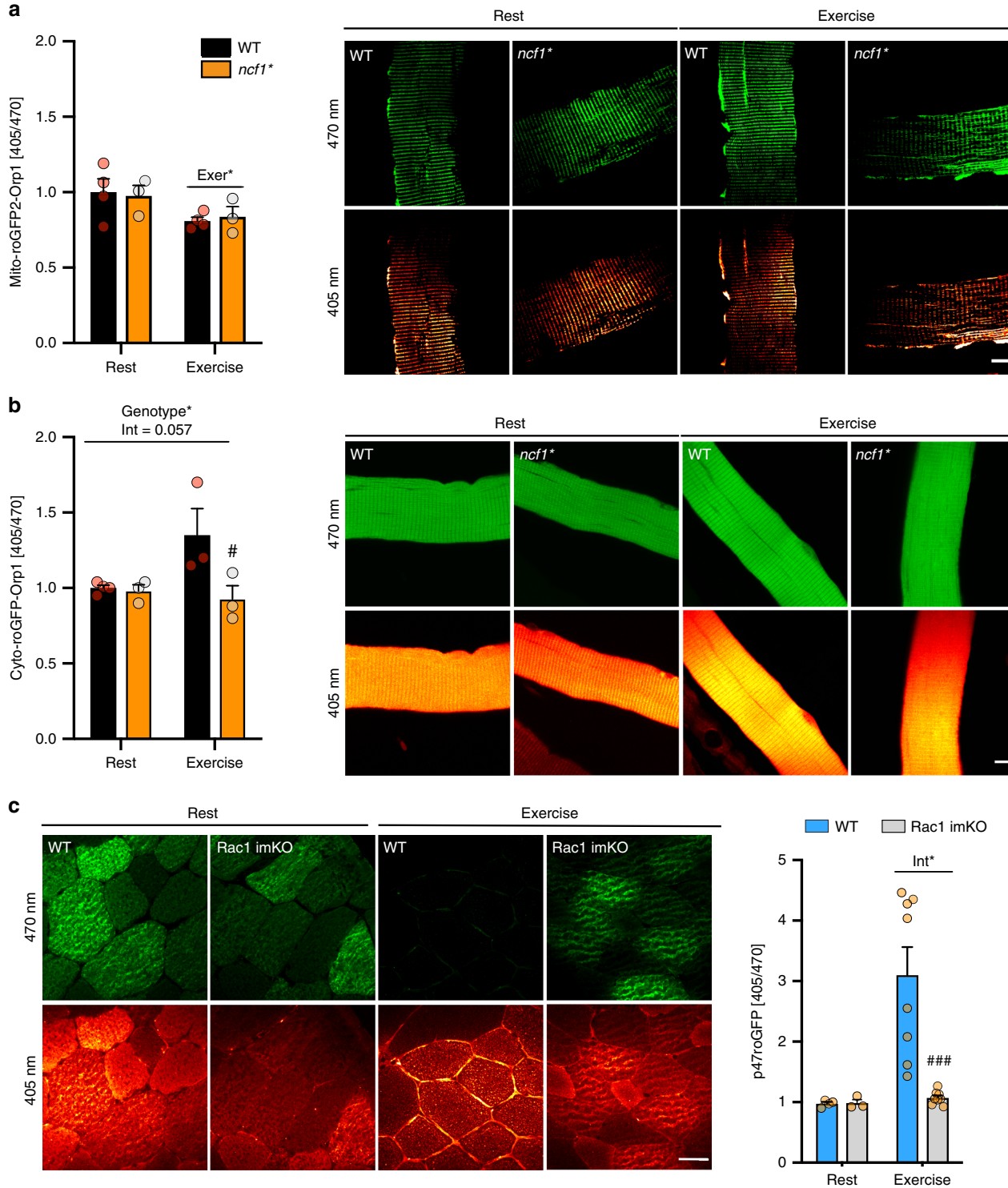

**Fig. 2** NOX2 is a major ROS source during exercise in skeletal muscle. Subcellularly targeted redox-sensitive GFP2 (roGFP2) were electroporated in *ncf1** and p47roGFP in inducible muscle-specific Rac1 mice. Representative image and quantification of **a** Mito-roGFP2-orp1 (WT, $n = 4$ and *ncf1** = 3), **b** cyto-roGFP2-Orp1 in Flexor Digitorum Brevis fibers (WT, $n = 3$–4 and *ncf1** = 3) and **c** p47roGFP oxidation in tibialis anterior muscle in WT and Rac1 imKO mice (WT, $n = 4/8$ for rest/exercise, Rac1 imKO, $n = 3/8$ for rest/exercise groups). Two-way ANOVA was performed to test for effects of exercise (Exer) genotype (Geno), and interaction (Int), followed by Tukey's post hoc test with correction for multiple comparisons. * Denotes $p < 0.05$ for main effects/ interaction. #, ### Denotes $p < 0.05$ and $p < 0.001$ compared to the WT group. Individual values and mean ± SEM are shown. Scale Bar = 10 μm (**a** and **b**)/30 μm (for **c**). For **a**–**c**, source data and *p* values are provided in the Source Data file

induced by exercise in concordance with previous in vitro studies[3,29,30].

We and others have shown that the small rho family GTPase Rac1 is necessary for insulin, contraction and passive stretch-induced skeletal muscle glucose uptake[13,14,31]. We have previously found this Rac1-dependency of glucose uptake and GLUT4 translocation to be more pronounced during in vivo treadmill exercise compared to in vitro contractions[17]. The

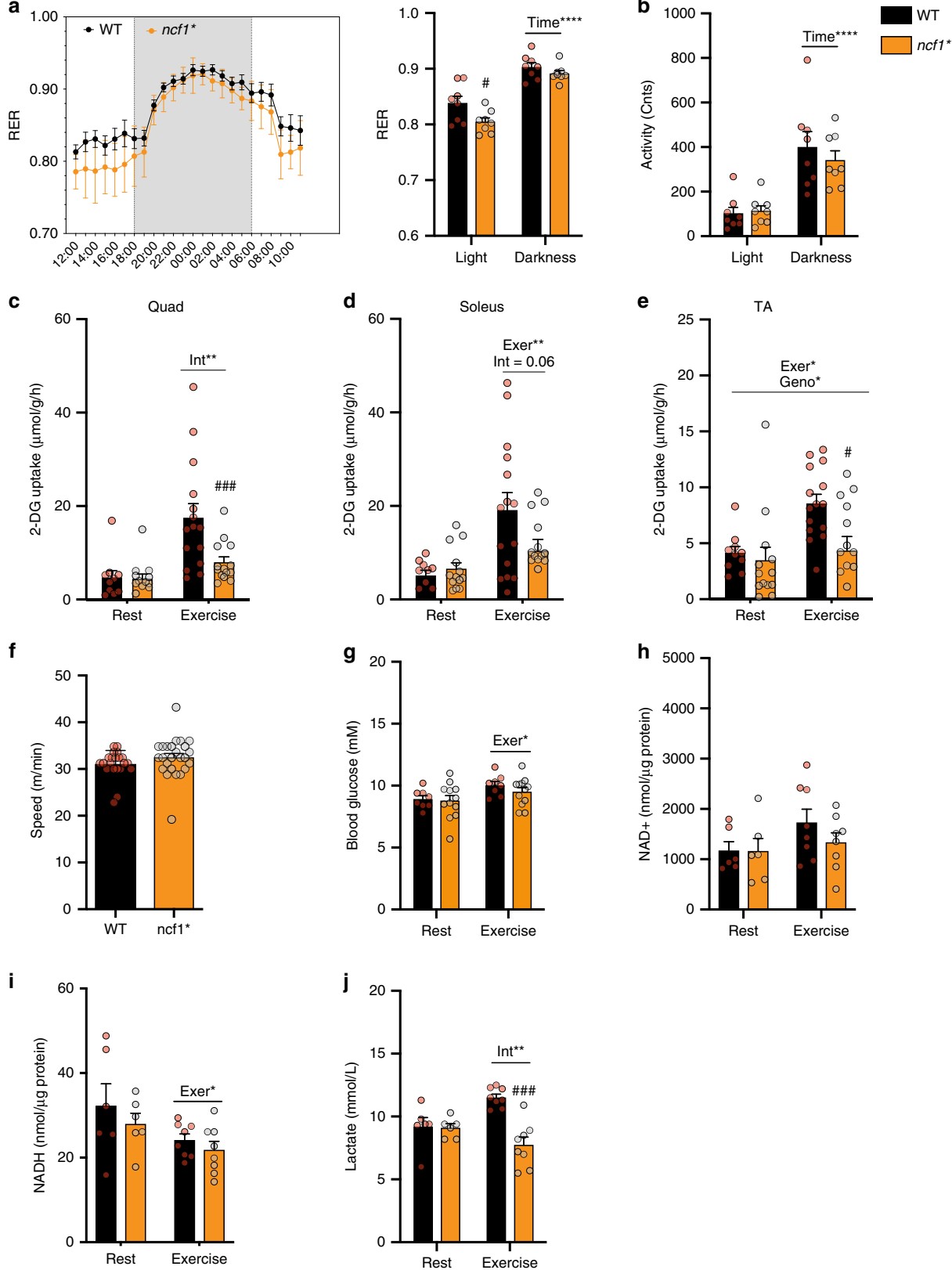

GLUT4 trafficking field has mostly focused on Rac1 as an orchestrator of the dynamic reorganization of the cortical actin cytoskeleton consisting of the cytoplasmic β and γ actin-isoforms[15]. This actin-reorganization was shown to be crucial for insulin-stimulated GLUT4 translocation in muscle cells[32]. However, the cytoplasmic β and γ actin-isoforms are both downregulated during muscle differentiation and whether dynamic reorganization occurs in adult skeletal muscle fibers similarly to in cells is presently unclear[33]. Our present data suggest that NOX2 is a vital downstream mediator of the effect of Rac1 on exercise-induced glucose uptake and GLUT4 transloca-tion, since the Rac1 imKO phenotype is shared by mice lacking

**Fig. 3** Exercise-stimulated glucose uptake is reduced in *ncf1\** mice. Different parameters were compared between p47phox-mutated mice (*ncf1\**) and wild-type (WT) mice. **a** Respiratory exchange ratio (RER) during light and dark cycles and **b** habitual activity (*n* = 8 per group). **c–e** In vivo running-induced (20 min, 65% maximal running speed) 2-deoxy-glucose uptake (2DG) uptake in quadriceps (Quad) soleus and tibialis anterior (TA) muscles from WT and *ncf1\** mice (*n* = 12–16). **f** Maximal running speed (*n* = 15 per group), **g** blood glucose concentration after rest/exercise (*n* = 8–12), **h** NAD$^+$, **i** NADPH levels in quadriceps muscle (*n* = 6–8 for rest/exercise groups), and **j** Plasma lactate concentration (*n* = 6–8 for rest/exercise groups). Unpaired *t*-test **f** and two-way ANOVA were performed to test the effects of exercise (Exer), genotype (Geno), and interaction (Int), followed by a Tukey's post hoc test with correction for multiple comparisons. *, **, **** Denotes $p < 0.05$, $p < 0.01$, $p < 0.0001$, respectively, for main effects/interaction. #, ## Denotes $p < 0.05$ and $p < 0.01$, respectively compared to the WT group. Individual values and mean ± SEM are shown. For all Items, source data and *p* values are included in the Source Data file

another essential regulatory subunit of NOX2, p47phox. In addition, our in vivo glucose uptake data provide a likely physiological mechanism explaining why exogenous antioxidants inhibit ex vivo contraction-stimulated and stretch-stimulated glucose uptake[10,11]. The relative dependence on actin and NOX2, and whether NOX2 and actin interact in muscle to regulate glucose uptake, as suggested in other cell types[34], should be clarified in future studies.

We observed no genotype difference in other possible determinants of muscle glucose uptake in vivo, including capillary density, muscle fiber-type, or protein expression of glucose metabolic proteins. consistent with our previous reports in *ncf1\**[35] and Rac1 imKO mice[17]. However, it remains possible that genotype-differences in glucose delivery to the muscle due to altered blood flow or intramyocellular glucose metabolism contributes to the currently observed phenotype. Technically challenging direct measurements of muscle blood flow and glucose metabolic flux in vivo will be required to investigate these possibilities in depth.

In the current study, the exercise-stimulated phosphorylation of p38MAPK was reduced in *ncf1\** mice compared to WT, in particular in quadriceps muscle. As p38 MAPK has been proposed as a regulator of muscular glucose uptake[11], we cannot exclude that reduced p38 MAPK contributed to the reduced glucose uptake we observed in *ncf1\** mice. However, the reduction in exercise-stimulated kinase signaling is less consistent than the reduction in glucose uptake in the different muscles analyzed in the *ncf1\** mice. Moreover, Rac1 imKO mice share the same glucose uptake phenotype without showing reductions in stretch-stimulated signaling, including p38 MAPK[17]. Instead, we speculate that the shared reduction in exercise-stimulated glucose uptake in *ncf1\** and Rac1 imKO mice might relate to a shared and yet undetermined redox-sensitive signaling mechanism. Worth noting, we believe NOX2 to signal independently of the AMPK-TBC1D1 signaling axis, based on the previous work by us and others[13,36–39]. In conjunction with the current study, strongly suggesting that Rac1 regulates glucose uptake via NOX2, we propose that NOX2 also signals independently of AMPK to regulate glucose uptake during moderate-intensity exercise. Future studies should work to identify the shared cell signaling traits between different NOX2-deficient mouse models using, e.g. redox proteomics[40].

In conclusion, this study showed for the first time that NOX2 is activated during moderate-intensity exercise in human and mouse skeletal muscle and is the primary source of ROS under such conditions. Furthermore, our comparison of two mouse models lacking regulatory NOX2 subunits showed that lack of ROS generation during exercise strongly impaired muscle glucose uptake and GLUT4 translocation during exercise. This indicates that NOX2 is a major source of ROS generation during exercise and that NOX2-dependent ROS production is an important signal for increasing muscle glucose uptake during exercise.

## Materials and methods

**Animals.** Male B10.Q WT and B10.Q. p47phox mutated (*ncf1\**) mice contain a point mutation in exon 8 generated in a previous study[20]. Age-matched WT and *ncf1\** mice between 12 and 16 weeks of age were used for experiments. Inducible muscle-specific male Rac1 mice (imKO) and littermate control mice were generated by crossbreeding Rac1$^{fl/fl}$ mice[41] with mice expressing Cre recombinase from a tetracycline-controlled transactivator coupled to the human skeletal actin promoter[42]. Control Rac1 WT mice were littermates carrying the Cre recombinase or the floxed Rac1 gene on none, one or both alleles. Rac1 mKO mice were homozygous for the floxed Rac1 gene and either homozygous or heterozygous for the Cre recombinase. Rac1 mKO was induced at 10–14 weeks of age by adding doxycycline in the drinking water (1 g/L; Sigma-Aldrich) for 3 weeks followed by a washout period of 3–5 weeks. Control wild type (WT) mice were littermates and also received doxycycline.

All mice were group-housed maintained on a 12:12-h light–dark cycle, at 21 °C, and received standard rodent chow diet (Altromin no. 1324; Chr. Pedersen, Denmark) and water ad libitum. All experiments were approved by the Danish Animal Experimental Inspectorate and complied with the "European Convention for the Protection of Vertebrate Animals Used for Experiments and Other Scientific Purposes."

**Human experiments.** Three young, healthy men (age 29 ± 3.56 years, subjects' characteristics are shown in Supplementary Table 2) gave their written, informed consent to participate in the study approved. The volunteers visited the laboratory on two separate days. The first day, the subjects completed an incremental test on a Monark Ergomedic 839E cycle ergometer (Monark, Sweden) to determine peak power output (PPO). On the second day, each subject performed a 30-min exercise trial at the intensity corresponding to ~65% of their individual peak power output in fasted state. Muscle biopsies were obtained from the m. vastus lateralis under local anesthesia [~3 ml xylocaine (20 mg ml$^{-1}$ lidocaine), Astra, Stockholm, Sweden] before and after exercise using a 5 mm Bergström needle with suction. The muscle biopsies were embedded in optimal cutting temperature compound (tissue-tek) and frozen in liquid nitrogen-cooled isopentane and stored at −80 °C for further analysis.

**Ethical regulations.** All experiments were approved by the Danish Animal Experimental Inspectorate (2015-15-0201-00477). Human experiments were approved by the Regional Ethics Committee for Copenhagen (H-16040740) and complied with the ethical guidelines of the Declaration of Helsinki II.

**In vivo gene transfer in adult skeletal muscle.** For the in vivo transfection experiments, mice were anesthetized with 2–3% isoflurane. Hyaluronidase (Sigma-Aldrich) dissolved in sterile saline solution (0.36 mg/ml) was injected subcutaneously on the plantar side of the foot near but not into the FDB and intramuscularly in TA muscle, followed by 20 and 40 μg plasmid injections 1 h later, respectively. FDB muscle electroporation was performed by delivering 10 electrical pulses at an intensity of 75 V/cm, 10-ms pulse duration, 200-ms pulse interval using acupuncture needles (0.20 × 25 mm, Tai Chi, Lhasa OMS) connected to an ECM 830 BTX electroporator (BTX Harvard Apparatus). TA electroporation was performed following the same protocol but raising the intensity to 100 V/cm (#45-0101 Caliper Electrode, BTX Caliper Electrodes, USA). The p47-roGFP2 construct used to determine NOX2 activity was a kind gift from Professor George G. Rodney[30]. The Mito-roGFP2-Orp1 (Addgene #64991) and Mito-roGFP2-Orp1 (Addgene #64992) plasmids were gifts from Professor Tobias Dick. The GLUT4-myc-GFP construct was a gift from Professor Jonathan Bogan (Addgene #52872). After electroporation, mice were allocated to their cages for 10 days before performing the experiments.

**Metabolic chambers.** O$_2$ uptake (VO$_2$) and CO$_2$ production (VCO$_2$) were measured using a CaloSys apparatus (TSE Systems, Bad Homburg, Germany). The data presented is the average for light and dark periods measured over 2 consecutive days in WT and *ncf1\** mice. The RER was calculated as VCO$_2$ production/VO$_2$ uptake.

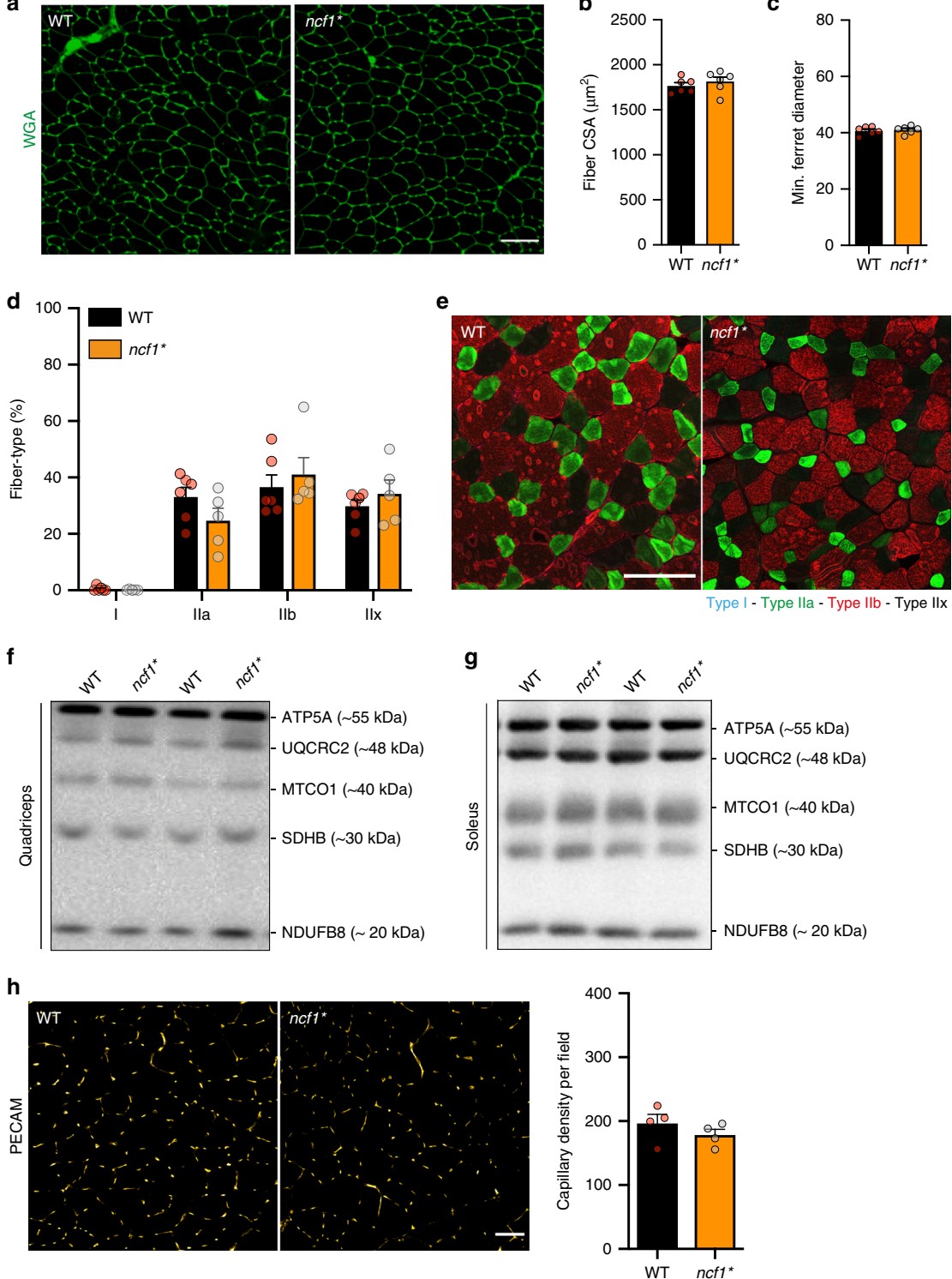

**Fig. 4** *Ncf1\** mice show similar muscle size, fiber-type composition, mitochondrial content and capillary density compared to WT group. **a** Fiber size-related parameters in WT and *ncf1\** mice were determined using wheat germ agglutinin (WGA) staining. **b** Myofiber cross-sectional area, and **c** minimum (min.) Ferret diameter. **d** Quantification and **e** representative merged image of fiber-type staining where Type I fibers (blue), Type IIa (green), Type IIb (red), Type IIx (non-stained) in WT ($n = 6$), and *ncf1\** ($n = 5$) TA muscles. Subunits of mitochondrial oxidative phosphorylation complexes were determined in **f** quadriceps and **g** soleus muscle lysates ($n = 14$ per group). **h** Capillary density was estimated by PECAM immunostaining in tibialis anterior sections. Two-way ANOVAs was performed to test for effects of exercise (Exer), genotype (Geno), and interaction (Int), followed by Tukey's post hoc test with correction for multiple comparisons. Individual values and mean ± SEM are shown. Scale Bar = 100 μm. For **b**, **c**, **d**, and **h**, source data and *p* values are provided in the Source Data file. For **f** and **g**, uncropped blots are provided in the Source Data file

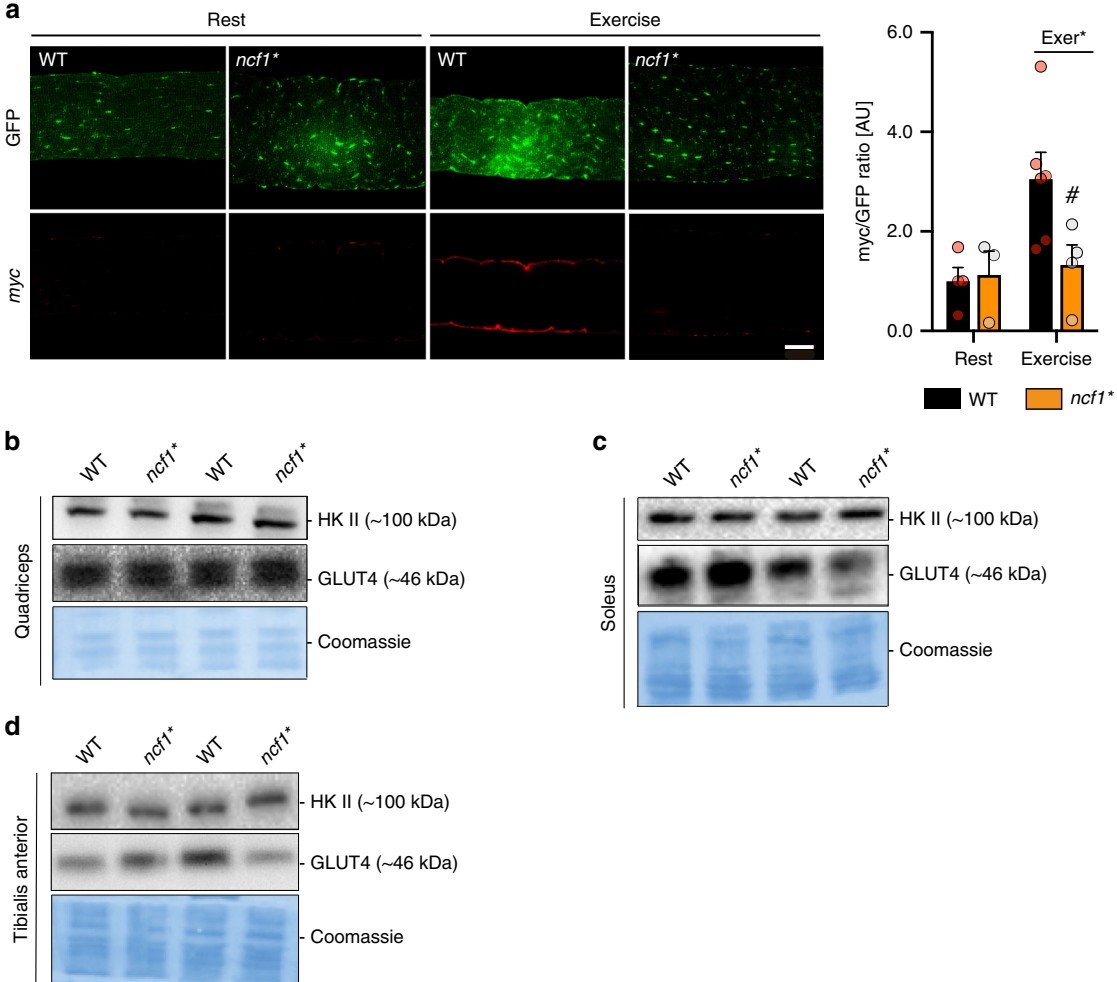

**Fig. 5** Exercise-stimulated GLUT4 translocation requires NOX2 activity. **a** the GLUT4-myc-GFP construct was electroporated into tibialis anterior muscles of both WT and *ncf1\** mice. Non-permeabilized muscle fibers from exercised (20 min, 65% maximal running speed) or resting mice were stained with anti-*myc* antibody and imaged by confocal microscopy ($n = 4$ mice, a minimum of 10 fibers/muscle). Total endogenous GLUT4 and HK II were determined by western blot in the following muscles **b** quadriceps, **c** soleus, and **d** TA muscles in WT and *ncf1\** mice ($n = 14$). Unpaired *t*-test (**b–d**) and two-way ANOVA were performed to test for effects of exercise (Exer), genotype (Geno), and interaction (Int), followed by a Tukey's post hoc test with correction for multiple comparisons. * Denotes $p < 0.05$ for main effect. # Denotes $p < 0.05$ compared to the WT group. Individual values and mean ± SEM are shown. Scale bar = 20 μm. For **a**, source data and *p* values are provided in the Source Data file. For **b–d**, uncropped blots are provided in the Source Data file

**Maximal running tests**. Mice were acclimated to the treadmill three times (15 min at 0.16 m/s) (Treadmill TSE Systems) a week prior to the maximal running tests. The maximal running test started at 0.16 m/s for 300 s with 10% incline, followed by a continuing increase (0.2 m/s) in running speed every 60 s until exhaustion. Exhaustion was defined as the point at which instead of running on the treadmill, the mice fall back on the grid three times within a 30 s period. Maximal running speed was determined as the last completed stage during the incremental test.

**FDB fiber isolation**. Muscle fibers were obtained by enzyme digestion of the whole muscle with collagenase type II (1.5 mg/ml) (Worthington, Lakewood, NJ) for 90 min at 37 °C followed by mechanical dissociation with fire-polished Pasteur pipettes. Isolated fibers were seeded in ECM Gel-coated (Sigma-Aldrich) cell culture dishes in DMEM supplemented with 10% fetal bovine serum. After 16 h of seeding, the fibers were used for experimentation.

**Live imaging**. FDB fibers were imaged in phenol red-free growth medium (5% horse serum) while kept at 95% $O_2$ and 5% $CO_2$ and 37 °C using a Pecon Lab-Tek S1 heat stage. Electrical stimulation protocol (10 V, 1 Hz, 0.1 ms duration) was delivered to isolated FDBs for 15 min via field electric field stimulation device (RC-37FS, Warner instruments, USA). Fibers were loaded with 20 nM tetramethylrhodamine, ethyl ester (TMRE+, Life Technologies) for 30 min before imaging.

**Exercise-stimulated GLUT4 translocation**. GLUT4 translocation was measured as recently described[43]. Briefly, rapidly dissected TA muscles were fixed by immersion in ice-cold 4% in paraformaldehyde in PBS for 4 h. TA muscles were transferred to a

Sylgaard dish and isolated into smaller bundles using fine forceps. The bundles were transferred to a storage solution containing PBS and glycerol (1:1) and kept at 4 °C overnight. Individual fibers were teased from fixed muscle with fine forceps under a dissection microscope. Non-permeabilized isolated muscle fibers were incubated in blocking buffer (1% BSA, 5% goat serum (16210-064, Gibco), 0.1% Na Azide (247-852, Merck) for 1 h and then incubated with an anti-*myc* antibody overnight (Supplementary Table I). The next day, fibers were then washed 3 × 10 min in PBS containing 0.04% saponin and incubated with Alexa 568 antibody (Supplementary Table I) in blocking buffer containing 0.04% saponin for 120 min Individual muscle fibers were transferred to mounting media (H-1000; Vector Laboratories) on a glass slide and covered by a coverslip. A minimum of 10 transfected fibers were blindly imaged from each muscle (a total of 60 fibers) using a ×63 1.4 NA Plan-Apochromat objective on a Zeiss 780 microscope driven by Zeiss Zen Black 2012. Image analysis is described in the image analysis sections and as previously described[43].

**Redox histology**. TA muscles were dissected and embedded in optimum cutting temperature (OCT) medium from Tissue-Tek, frozen in melting isopentane and kept at −80 °C until processing. An independent set of TA muscles were used for western blotting. Total oxidant levels were estimated as previously described[12]. Briefly, 10 μm thickness muscle cryosections were incubated with 5 μM of 2′,7′-DCFH (Molecular Probes, Eugene) and allowed to dry overnight at room temperature in dark. Muscle membranes were visualized using Texas red-labeled wheat germ agglutinin (WGA; Molecular Probes). The redox histology was performed as previously described[18,44] and in Supplementary Fig. 3A. Whole FDB muscles were immediately immersed in PBS containing freshly dissolved NEM (100 mM) in ice and then fixed in 4% PFA in PBS (pH 7.4) plus 100 mM NEM for 2 h at 4 °C.

p47roGFP-transfected TA cryosections were incubated with PBS containing 50 mM N-ethylmaleimide (NEM) for 10 min at 4 °C, followed by fixation using 4% paraformaldehyde for 10 min at room temperature. All samples were mounted in mounting medium (H-1000; Vector Laboratories). Imaging is described in the "Imaging and image analysis" section.

**Cryosection immunostaining**. Cryosections of TA muscle (cut in transverse orientation) were stained with monoclonal anti-myosin heavy chain (MyHC) antibodies (DSHB, University of Iowa): BA-D5 (IgG2b, supernatant, 1:100 dilution) specific for MyHC-I, SC-71 (IgG1, supernatant, 1:100 dilution) specific for MyHC-2A, and BF-F3 (IgM, purified antibody, 1:100 dilution) specific for My-HC-2B. Type 2X fibers are not recognized by these antibodies, and so appear black. Three different secondary antibodies (Life Technologies, Carlsbad, USA) were used to selectively bind to each primary antibody: goat-anti-mouse IgG2b, conjugated with Alexa 647 fluorophore (to bind to BA-D5); goat-anti-mouse IgG1, conjugated with Alexa 488 fluorophore (to bind to SC-71); goat-anti-mouse IgM, conjugated to Alexa 555 fluorophore (to bind to BF-F3). Muscle sections, 10 μm thick, were fixed in 2% PFA for 30 min and permeabilized with 0.1% Triton-X100 for 10 min After washing in immunobuffer (IB = 0.25% BSA, 50 mM glycine, 0.033% saponin, and 0.05% sodium azide diluted in PBS), sections were blocked for 1 h in 2% BSA, then briefly washed twice with IB for 5 min A solution with all the primary antibodies diluted in IB was then prepared, and sections were incubated for 2 h at 37 °C. After three washes (10 min each) with IB, sections were incubated for 1 h at 37 °C with a solution containing the three different secondary antibodies diluted in IB. After three washes with IB (10 min each) and a brief rinse in PBS, sections were mounted with Fluoromount (Sigma-Aldrich, St. Louis, USA). Capillary density was evaluated using a PECAM1 immunostaining in TA cryosections. After fixation and blocking steps as described above, anti-PECAM1 antibody (SCBT, M-20) diluted 1:100 in 1% BSA was incubated overnight at 4 °C. After three washes (5 min each), an anti-goat Alexa 488-conjugated secondary antibody was incubated for 1h. After three washes with IB (5 min each) sections were mounted.

**Imaging and image analysis**. For all live-imaging experiments, confocal images were collected using a ×63 1.4 NA oil immersion objective lens on an LSM 780 confocal microscope (Zeiss) driven by Zen 2011. TMRE$^+$ fluorescence was detected using the excitation-emission λ545–580/590 nm. For the roGFP biosensors images, raw data of the λ405-nm and 488-nm laser lines were exported to ImageJ as 16-bit TIFFs for further analysis[44]. For visualization of the ratiometric images, the ImageJ lookup tables "Green" and "Red Hot" were used.

Data are presented as fluorescence ratio (λ405/488 nm) normalized to the resting WT group. For fiber-type immunostainings, images were collected using a dry ×20 0.8 NA Plan Apo objective on an LSM 710 confocal microscope (Zeiss) driven by Zen 2012. The used excitation laser lines were 488, 561, and 633 nm, respectively, assembled by Zeiss. Three tracks were sequentially used for acquisition, with 488 and 633 channels recorded with PMTs, while the 561 channel was recorded with a GaAsP detector array. The matching dichroic mirrors were used for all channels, and the pinhole was set at 1 AU for 580 nm. Images were exported to ImageJ and the composition of different MHC fiber types were quantified by counting sections of each muscle bed (Type I, blue; Type IIa, green; Type IIb, red; Type IIx, black) using the Cell Counter plugin. To visualize membrane GLUT4, both *myc* and GFP images were duplicated and the new images underwent a background subtraction, a gaussian blur adjustment, and were converted to binary images. From these binary images, regions of positive GFP and *myc* signal were selected and overlaid onto the original images for quantification of integrated density of the GFP and *myc* signal, respectively. Quantification were expressed as the ratio of surface *myc* signal to total cellular GFP fluorescence for each fiber.

**Running-stimulated muscle 2-deoxyglucose (DG) uptake**. Muscle-specific 2DG uptake was measured as previously described[36]. [³H]2DG (PerkinElmer) was injected intraperitoneally (as a bolus in saline, 10 μL/g body weight) containing 0.1 mmol/L 2DG and 50 μCi/mL [³H]2DG corresponding to ~12 μCi/mouse) into fed mice immediately before the exercise bout. Blood glucose was measured before and immediately after 20 min of exercise. Mice were euthanized by cervical dislocation, and muscle tissues and plasma samples quickly frozen in liquid nitrogen and stored at −80 °C until analysis. Samples were deproteinated using 0.1 mol/L Ba (OH)$_2$ and 0.1 mol/L ZnSO$_4$. The total muscle [³H]2DG tracer activity found in 2DG-6-phosphate was divided by the area under the curve of the specific activity at time points 0 and 20 min multiplied with the average blood glucose at time points 0 and 20 min This was related to muscle weight and the time to obtain the tissue-specific 2DG uptake as micromoles per gram per hour.

**Western blot analyses**. Tissue was homogenized for 1 min at 30 Hz using a Tissue Lyser in ice-cold lysis buffer (0.05 mol/L Tris Base pH 7.4, 0.15 mol/L NaCl, 1 mmol/L EDTA and EGTA, 0.05 mol/L sodium flouride, 5 mmol/L sodium pyrophosphate, 2 mmol/L sodium orthovanadate, 1 mmol/L benzamidine, 0.5% protease inhibitor cocktail (P8340, Sigma Aldrich), and 1% NP-40). After rotating end-over-end for 30 min, lysate supernatants were collected by centrifugation (18,327 × g) for 20 min at 4 °C. Lysate protein concentrations were determined using BSA standards and bicinchoninic acid assay reagents (Pierce). Total protein and

phosphorylation levels of relevant proteins were determined by standard immunoblotting techniques, loading equal amounts of protein.

The primary antibodies used p-AMPK$^{Thr172}$ diluted 1:1000 in 3% BSA (Cell Signaling Technology (CST)), #2535s), p-p38 MAPK$^{Thr180/Tyr182}$ diluted 1:1000 in 3% BSA (CST, #9211), Erk1/2$^{Thr202/Tyr204}$ diluted 1:1000 in 3% BSA (CST, #9101), p-ACC2 Ser$^{212}$ diluted 1:1000 in 3% BSA (Millipore, 03-303), GLUT4 (ThermoFisher Scientific, PA-23052), Rac1 diluted 1:1000 in 2% skim milk (BD Biosciences, #610650), NOX2 diluted 1:1000 in 2% skim milk (Abcam, #Ab129068), Catalase diluted 1:750 in 2% skim milk (SCBT, sc-271803), MnSOD diluted 1:1000 in 2% skim milk (Millipore, 06-984), TRX2 diluted 1:1000 in 2% skim milk (SCBT, sc-50336), actin diluted 3:1000 in 2% skim milk (CST, #4973) total p38 MAPK diluted 1:000 in 2% skim milk (CST, #9212), α2 AMPK diluted 1:1000 in 2% skim milk (a gift from D. Grahame Hardie, University of Dundee), total ERK 1/2 diluted 1:1000 in 2% skim milk (CST, #9102), and TBC1D1$^{ser231}$ diluted 1:1000 in 3% BSA (Millipore #07-2268), OXPHOS cocktail diluted 1:5000 in 3% BSA (Abcam, #ab110413), total TBC1D1 diluted 1:1000 in 2% skim milk (Abcam, #Ab229504), and Hexokinase II diluted 1:1000 in 3% BSA (CST, #2867). ACC protein was detected using horseradish peroxidase-conjugated streptavidin from Dako (P0397), dilution 1:3000 in 3% BSA (Supplementary Table 1). All antibodies were optimized for signal linearity. Membranes were blocked for 1 h at room temperature in TBS-Tween 20 containing either 2% skimmed milk or 2% BSA and incubated overnight at 4 °C with primary antibody. The membranes were incubated in the corresponding horseradish peroxidase-conjugated secondary antibody for 1 h at room temperature and washed in TBS-T before visualization of the proteins (ChemiDoc$^{TM}$MP Imaging System, Bio Rad). Uncropped blots from the main Figs. are shown in the Source data File.

**Statistical analyses**. Results are shown as individual values and mean ± S.E.M. Statistical testing was carried out using *t*-tests or two-way (repeated measures when appropriate) ANOVA as applicable. Tukey's post hoc test was performed for multiple comparisons when ANOVA revealed significant main effects or interactions. Statistical analyses were performed using GraphPad Prism v8. All *p* values are provided in the Source Data File.

**Reporting summary**. Further information on research design is available in the Nature Research Reporting Summary linked to this article.

## Data availability
The authors declare that the data supporting the findings of this study are available within the Article, Supplementary Information files, and Source Data, or are available upon reasonable requests to the authors. The source data underlying Figs. 1a–d, 2a–c, 3a–j, 4b, c, d, f–h, and 5a–d are provided as a Source Data file.

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

## Acknowledgements

T.E.J. was supported by a Novo Nordisk Foundation Excellence project grant (#15182). C.H.-O. was supported by a Chilean CONICYT Ph.D. Scholarship. Ph.D. stipends to J.R.K. were co-funded by the Danish Diabetes Academy, funded by the Novo Nordisk Foundation. Z.L. was supported by Chinese Scholarship Council Ph.D. stipends. L.S. was supported by a Danish Research Council grant 4004-00233 and Novo Nordisk Foundation Excellence grant NNF18OC0032082. E.J. was supported by FONDECYT grant 1151293. E.A.R. was supported by the Danish Council for Independent Research/Science 6108-00203. R.H. was supported by the Swedish Research Council, the Knut and Alice Wallenberg foundations and the Swedish Cancer Society. We thank Kim Anker Sjøberg for pre-testing the human subjects and Betina Bolmgren for technical assistance with measuring 2DG uptake. Imaging data were collected at the Center for Advanced Bioimaging and the Core Facility for Integrated Microscopy, University of Copenhagen, Denmark. Illustrations of Supplementary Fig. 3A and were created using BioRenders (BioRenders.com).

## Author contributions

Conceptualization, C.H.-O., E.J., and T.E.J. Methodology, C.H.-O. and T.E.J. Investigation, C.H.-O., J.R.K., Z.L., E.D., J.T.T., S.H.R., R.H., E.A.R., and L.S. Formal analysis, C.H.-O. Visualization, C.H.-O. Writing—original draft, C.H.-O. and T.E.J. Writing—review and editing, C.H.-O. with input from all authors. Funding acquisition, T.E.J. Supervision, E.J. and T.E.J.

## Competing interests

The authors declare no competing interests.
