## [Peer Review File · Nature Communications]

Reviewers' Comments:

Reviewer #1:

Remarks to the Author:

The authors combined exercise in humans and mice with fluorescent dyes, genetically-encoded biosensors, and NADPH oxidase 2 (NOX2) loss-of-function models to demonstrate that NOX2 is the main source of ROS during moderate-intensity exercise in skeletal muscle.

Despite the elegant methodology of the present work, the current finding does not reveal any major novel concepts of the GLUT4 translocation mechanisms. The evidence presented in the manuscript is rather preliminary and not entirely convincing. However, the study contains interesting and important new details concerning signalling events mediated by muscle contractions. Thus, there are several important issues that should be addressed.

General comments:

The mechanistic link between NOX2 and GLUT4 translocation machinery is missing. The ROS-activated molecular trigger for GLUT4-contained vesicles movement is not established. NOX derived superoxide has been shown to enhance mitochondrial ROS production (Ahmed et al 2011, *Biochem. J.*, PMID 21967515). It is possible that within the current constructs, both in Rac1 and nfc1 deficient mice the mitochondrial ROS production is also decreased, thus contributing to the reduction in Glut4 translocation and ensuing decrease in glucose uptake.

ACC phosphorylation is used as a proxy of AMPK activity. However, phosphorylation depends also on protein phosphatase 2 activity, and thus AMPK activity could be underestimated. Considering this, few other pathways might be worth investigating.

The authors list that TBC1D1 is not likely to contribute, but however this would be easy to examine by assessing exercise responsive phosphorylation sites by Western blot. Ca²⁺ release, SERCA activation and the ensuing increase in AMP/ATP ratio are suspected to play a role in Glut4 translocation in response to contraction. It might be of interest to examine whether ER or SERCA regulation is affected in nfc1 and rac1 deficient mice (i.e. PKCs or CaMKII) .

Considering the decrease in fat mass, and increased in lean mass in nfc1* deficient mice, have you measured average fibre cross sectional area? Additionally, it would be of interest to determine the specific fibre type comprising the muscles.

Methodological considerations:

As the authors state, DCFH is an unspecific detector of cellular ROS. The mitochondria are no longer considered to be the primary source of ROS in exercise. However, the decrease in NOX2 activity could possibly lead to decreased ROS production from the mitochondrial sources. If it is feasible, utilizing a mitochondrial specific dye, such as MitoSOX in in vitro cells might be interesting to further discern where the changes in ROS are only due to lower NOX2 contribution.

Even though the p47phox-roGFP constructs are referenced in the methods, it might be of interest to briefly explain their sensitivity to the redox status.

Minor comments:

The DCFH oxidation in nfc1 deficient mice in the basal state seems to be increased (although not significantly). Have the authors any speculation on where this increase is coming from? In figures utilizing p47-roGFP construct, would it be feasible to visualize the membrane and show an overlay of the signals to make the translocation of the p47phox clearer?

Fig 1A – Was the parametric t-test used to compare fold change or overall values?

Fig 1B – No total blots for p-38 MAPK, ERK, TBC1D1 and ACC. This information should not be even Supplementary figures.

Fig 1D – Very hard to discern the 405nm signal. Could benefit from visualizing the membrane and overlaying the signals or changing the contrast.

Fig 2I – Was there no overall effect shown in the ANOVA?

Fig3 E and F – Two complexes are marked as Complex IV, and there is no Complex V.

Fig4 A, B and C – No total ACC blot present.

Fig S2C - There is no Rac1 DCF staining in Rest condition.

Reviewer #2:

Remarks to the Author:

The major claims of the paper are that NOX2 is a major muscle source of ROS that regulates muscle GLUT4 translocation and glucose uptake during moderate intensity exercise, and that the role of Rac1 in regulating these processes are due to Rac1's essential role in NOX2 activation. These are novel claims that will be interesting to researchers interested in the regulation of muscle glucose uptake with exercise, and likely others more generally interested in glucose metabolism. The results with DCF in exercised muscles from WT compared to *ncf1** mice are consistent with the interpretation that NOX2 is important for ROS production during exercise. These results have the potential to influence thinking in the field. The level of influence will depend on providing a convincing explanation about the results in Figure 1 as discussed below.

Fig 1A (exercise by human) and Fig1C (exercise by mouse) look quite different. In the human exercise section, there is uniform signal across all fibers and within all fibers. In the mouse exercise section, there is non-uniform signal both within and across fibers. Some fibers have no signal, and the fibers with more signal appear to have discrete hotspots of signaling that are largely at the outer edge of fibers. There should be a discussion and possible explanation regarding the species differences, and also regarding the differences in subcellular regional differences in mouse fibers, and between mouse fibers. Have sections been stained to identify muscle fiber type to see if this is related to the heterogeneity in signaling in mouse fibers, and the uniformity in human fibers? Including information on the fiber type and also on measurement of changes in glycogen content in serial sections would be informative.

For Figure 2C and 2D, it is surprising that 2DG uptake is very similar between the QUAD and soleus. Higher values would be expected both at rest and with exercise in the soleus. What accounts for the unexpected results?

The in vivo glucose uptake during exercise will be greatly influenced by altered blood flow. Could the results in the KO models be related in part to altered blood flow?

Lines 235-6 refer to previous work that provided evidence that NOX2 signals independently of AMPK-TBC1D1, but reading the cited work, the evidence supporting this conclusion is not clear, and the text should more specifically identify the experimental evidence that is the basis for this statement.

Citation #31 states that Rac1 accounts for the majority of muscle glucose uptake with ex vivo contraction, but not in vivo exercise. This result appears to be in conflict with the current results. Please clarify.

Lines 175-6 states that exercise-induced ROS production is suggested to activate a number of kinases linked to glucose uptake-regulation in muscle based on citations #23. However, looking at that citation, it is not obvious which specific kinases are being referred to. Please clarify.

Citation #23 claims that "exercise at moderate intensity (50–75%VO₂max), the primary source of ROS is from the mitochondria." Please clarify how this statement corresponds to the results in this study.

For the purpose of assisting with researchers who wish to reproduce the work, some minor sources of confusion include the description on line 263 of "injected subcutaneously in FDB." Was the solution actually injected into the FDB itself subcutaneously near the FDB? On lines 370-2, there is description of freezing tissue embedded in tissue-tek. Were these samples used for western blotting?

Line 602 – please define IQR.

Reviewer #3:

Remarks to the Author:

This is a potentially important paper describing the role of NADPH oxidase 2 in skeletal muscle glucose uptake in rodents. The data presented are generally compelling and the use of specific knockout models provides reassurance on the in vivo biological relevance of the findings.

There are however some substantial concerns with the methods used in one part of the manuscript, specifically the assessments of muscle ROS production and NADPH oxidase activation:

The authors have assessed muscle ROS production ex vivo by assessment of DCF oxidation (presumably they used the reduced form, usually described as DCFH) in cryostat sections of human muscle biopsies or TA muscles from mice. Muscle sections were incubated with DCFH and allowed to dry overnight at room temperature. While this method has been previously published, the assay is not validated and it is not clear what is being measured or where the species that oxidise the DCFH derive from. Thus ROS (superoxide) generation during contractile activity of muscle occurs very rapidly and this species and reactive species derived from them degrade very rapidly. Since the assay is performed on muscle ex vivo, this assay cannot be directly measuring ROS generation that has occurred in vivo. Furthermore the sections are effectively incubated with the DCFH over a period up to 24 hours which allows a great deal of time for spurious DCFH oxidation deriving from many sources including (for instance) iron delocalisation within the cryosections. It may be that the assay can provide some measure of the susceptibility of the muscle sections to oxidation, but it cannot be an assay for ROS generation in muscle during exercise.

The assay for activation of Nox2 enzymes based on oxidation of p47-roGFP is novel and based on a recently described histological approach to examine the redox status of ro-GFP based probes. However this approach does not appear to have been applied to the p47-roGFP probe previously and hence the reliability and validity of the assay has not been established. Some validation of the assay is required. It is also apparent that there is heterogeneity of fluorescence between fibres (how is this dealt with in the quantification) and the pattern of fluorescence seen in the images is not reflected in the summary graphs.

Reviewer #4:

Remarks to the Author:

In this study, investigators set out to examine the role of NOX2 in generating ROS in skeletal muscle during moderate exercise, and importance for glucose metabolism.

Overall this paper is extremely well executed in terms of directness of addressing the research question, adequate experiments, intuitive and well designed figures, as well as a very good text flow and excellent lingual command.

Abstract and introduction is clear and well written. The investigation of the function of ROS during exercise in skeletal muscle is very interesting, and as the authors mention, underinvestigated

previously.

Methods.

The authors start by framing the research question in an investigation using muscle biopsies from healthy volunteers (n=3, in this case data are sufficiently clear to prove the point).

Murine models and gene transfer. Inducible muscle-specific male Rac1 mice and littermate control mice were used, with inducible gene expression under doxycyclin control. This nicely produces both the experimental group and the controls. Electroporation was used for gene transfer which is a clever choice for this study. However use of hyaluronidase seems unnecessary (electroporation alone has been deemed sufficient to obtain high level gene transfer). The amount of DNA is very high (20 or 40 μg for the tibialis anterior muscle). 5 μg (or even less) should have been sufficient. It is not clear if the authors performed empty vector controls, which should be included. It is not stated how much time passed from the gene electrotransfer to experiments. Statistical methods in order and appropriately described.

Figures

Over all very well laid out and easily interpretable. Microscopy clear.

Minor comments:

Fig 1. By indicating 'human' and 'murine' eg by the x-axis this important information would be legible in the figure.

Fig 2a. X-axis labels not intuitive (why the .00 ?) the legend is more appropriately placed in the right hand corner - not in different colored sections of the graph.

Minor

Line 197 - why the comma? Insulin-stimulated, muscle contraction

Reviewer #1 (Remarks to the Author):

The authors combined exercise in humans and mice with fluorescent dyes, genetically-encoded biosensors, and NADPH oxidase 2 (NOX2) loss-of-function models to demonstrate that NOX2 is the main source of ROS during moderate-intensity exercise in skeletal muscle.

Despite the elegant methodology of the present work, the current finding does not reveal any major novel concepts of the GLUT4 translocation mechanisms.

We find this comment somewhat odd and harsh.

We fail to understand how linking GLUT4 translocation and glucose uptake to NOX2 during physiologically relevant *in vivo* exercise lacks novelty given that the ROS source increasing muscle glucose uptake was previously unknown and Rac1 was previously believed to act mainly through actin.

The evidence presented in the manuscript is rather preliminary and not entirely convincing. However, the study contains interesting and important new details concerning signalling events mediated by muscle contractions. Thus, there are several important issues that should be addressed.

General comments:

The mechanistic link between NOX2 and GLUT4 translocation machinery is missing. The ROS-activated molecular trigger for GLUT4-contained vesicles movement is not established.

Thanks for pointing this out. Respectfully, we think that identifying the molecular link between NOX2 and GLUT4 lies well beyond the scope of this paper because a) we do have not any obvious redox-sensitive protein candidates to test and b) redox-proteomics in WT vs. NOX2-deficient mice to identify and subsequently characterize such candidates would realistically take another 3-4 years.

NOX derived superoxide has been shown to enhance mitochondrial ROS production (Ahmed et al 2011, Biochem. J., PMID 21967515). It is possible that within the current constructs, both in Rac1 and *nfc1* deficient mice the mitochondrial ROS production is also decreased, thus contributing to the reduction in Glut4 translocation and ensuing decrease in glucose uptake.

The reviewer raises an important point about the potential crosstalk between different subcellular ROS sources, a topic which is completely unstudied in exercising muscle. To address this concern, we have now completed additional experiments electroporating WT and *ncfl** with mitochondrially targeted roGFP2-orp1 and cytosolic roGFP2-orp1 to be able to measure compartmentalized hydrogen peroxide production during *in vivo* running in WT vs. *ncfl** mice. These data are included in Fig. 2 and show no genotype difference in mitochondrial ROS production at base-line and a similar effect of exercise to reduce mitochondrial ROS production in both genotypes. The cytosolic roGFP-Orp1 probe oxidation showed a genotype-difference, increasing with exercise in WT but not *ncfl** muscle fibers, supporting that NOX2 but not mitochondria is the major ROS source during moderate intensity *in vivo* exercise, in good overall agreement with previous *in vitro* studies.

ACC phosphorylation is used as a proxy of AMPK activity. However, phosphorylation depends also on protein phosphatase 2 activity, and thus AMPK activity could be underestimated.

Considering this, few other pathways might be worth investigating.

The authors list that TBC1D1 is not likely to contribute, but however this would be easy to examine by assessing exercise responsive phosphorylation sites by Western blot. Ca²⁺ release, SERCA activation and the ensuing increase in AMP/ATP ratio are suspected to play a role in Glut4 translocation in response to contraction. It might be of interest to examine whether ER or SERCA regulation is affected in *nfc1* and *rac1* deficient mice (i.e. PKCs or CaMKII).

Thanks, we have now measured SERCA1, EF2 Thr57 phos, CaMKII Thr286/287 phos, and TBC1D1 Ser231 phos after *in vivo* exercise in quad muscle. None of these change in the *in vivo* samples, likely because of the time needed to collect the muscles after *in vivo* exercise and/or the moderate intensity exercise. We present the data to the reviewers below. Since they do not show any difference, we have decided not to put these in the manuscript and just state that we found no difference in these. However, if the reviewer finds that these data should be included in the manuscript as a supplemental figure that can be done.

Considering the decrease in fat mass, and increased in lean mass in *nfc1** deficient mice, have you measured average fibre cross sectional area? Additionally, it would be of interest to determine the specific fibre type comprising the muscles.

Excellent questions. In fig. 4, we have now measured and included Fiber CSA, minimum Ferret Diameter and fiber-type data in *Ncf1** mice and see no genotype-differences in these parameters. Similar results were previously obtained in *Rac1* deficient mice.

Sylov (2017) Diabetes 66:1548–1559

Methodological considerations:

As the authors state, DCFH is an unspecific detector of cellular ROS. The mitochondria are no longer considered to be the primary source of ROS in exercise. However, the decrease in NOX2 activity could possibly lead to decreased ROS production from the mitochondrial sources. If it is feasible, utilizing a mitochondrial specific dye, such as MitoSOX in *in vitro* cells might be interesting to further discern where the changes in ROS are only due to lower NOX2 contribution.

Interesting proposal which inspired us in part to perform the aforementioned mitochondrial roGFP and cytosolic roGFP oxidation in WT vs. *ncfl** mice after *in vivo* running.

As stated, mito-roGFP decreases genotype-independently with exercise whereas cyto-roGFP shows a genotype-difference increasing in WT but not *ncfl** mice, supporting the DCFH data.

Even though the p47phox-roGFP constructs are referenced in the methods, it might be of interest to briefly explain their sensitivity to the redox status.

Thanks, we added a paragraph in the results elaborating on how the redox-sensitive roGFP is believed to work. Additionally, we included a cartoon describing the *in vivo* redox-preservation method in Fig. S3A.

Minor comments:

The DCFH oxidation in *ncfl* deficient mice in the basal state seems to be increased (although not significantly). Have the authors any speculation on where this increase is coming from?

This increase was not significant, so we chose not to speculate about it in the paper. Since baseline fat oxidation is increased in *ncfl** vs. WT and associated with increased mitochondrial ROS production (Anderson, Lustig et al. 2009, Fukawa, Yan-Jiang et al. 2016), we speculated that this was a possible explanation. However, the mitochondrial and cytosolic roGFP2-Orp1 data suggest that this may not be the case.

Since we do not know what the DCFH oxidation signal reflects in the muscle, we do not have a good explanation at the moment.

In figures utilizing p47-roGFP construct, would it be feasible to visualize the membrane and show an overlay of the signals to make the translocation of the p47phox clearer?

Interesting proposal which we discussed at length. We actually do not see a clear change in p47-roGFP-pattern upon exercise. This is likely because only a small fraction of p47roGFP translocates and NOX2 resides not only in the sarcolemma but also in t-tubules in muscle and perhaps in various endomembrane-compartments based on work in other cell-types. Hence, we do not believe that this set of experiments would have the sensitivity to demonstrate an *in vivo* exercise-stimulated increase in p47phox translocation.

Fig 1A – Was the parametric t-test used to compare fold change or overall values?

Thank you for pointing out this lack of clarity. We ran a paired T-test on the absolute values, but the graph depicts the relative increase to illustrate that all subjects increase. To avoid misunderstanding we now show the absolute values in Fig. 1A.

Fig 1B – No total blots for p-38 MAPK, ERK, TBC1D1 and ACC. This information should not be even Supplementary figures.

We have now measured and included the total proteins for the 3 human subjects in Fig. S1A. As expected, none of these changed with a single bout of exercise.

Fig 1D – Very hard to discern the 405nm signal. Could benefit from visualizing the membrane and overlaying the signals or changing the contrast.

We have now changed the pseudo color of the 405nm panels to increase the visibility.

Fig 2I – Was there no overall effect shown in the ANOVA?

Yes, this mistake is now corrected now. Thanks.

Fig3 E and F – Two complexes are marked as Complex IV, and there is no Complex V.

Annotation corrected, thanks.

Fig4 A, B and C – No total ACC blot present.

We have now compared total protein levels between WT and *ncf1** mice for AMPK alpha2, total p38 MAPK, total ERK and total ACC and see no difference in any of these. Included in Fig. S2.

Fig S2C - There is no Rac1 DCF staining in Rest condition.

For breeding-reasons, we prioritized statistical power in the exercise group in the original experiment. However, we have now ascertained this in a separate experiment and include a separate

supplemental panel with the rest condition genotype comparison. Moreover, our p47roGFP biosensor data suggest no differences in NOX2 activity at rest between Rac1 and WT mice.

Reviewer #2 (Remarks to the Author):

The major claims of the paper are that NOX2 is a major muscle source of ROS that regulates muscle GLUT4 translocation and glucose uptake during moderate intensity exercise, and that the role of Rac1 in regulating these processes are due to Rac1's essential role in NOX2 activation. These are novel claims that will be interesting to researchers interested in the regulation of muscle glucose uptake with exercise, and likely others more generally interested in glucose metabolism. The results with DCF in exercised muscles from WT compared to *ncf1** mice are consistent with the interpretation that NOX2 is important for ROS production during exercise. These results have the potential to influence thinking in the field. The level of influence will depend on providing a convincing explanation about the results in Figure 1 as discussed below.

Fig 1A (exercise by human) and Fig1C (exercise by mouse) look quite different. In the human exercise section, there is uniform signal across all fibers and within all fibers. In the mouse exercise section, there is non-uniform signal both within and across fibers. Some fibers have no signal, and the fibers with more signal appear to have discrete hotspots of signaling that are largely at the outer edge of fibers. There should be a discussion and possible explanation regarding the species differences, and also regarding the differences in subcellular regional differences in mouse fibers, and between mouse fibers. Have sections been stained to identify muscle fiber type to see if this is related to the heterogeneity in signaling in mouse fibers, and the uniformity in human fibers? Including information on the fiber type and also on measurement of changes in glycogen content in serial sections would be informative.

Interesting proposition, which made us look though our previous DCFH stainings in mice. Basically, the mouse stainings shown are not very representative and we have also produced mouse TA stainings that look more similar to humans (see below, left). We include some examples below from other recent measurements in independent projects in mouse muscle (below, left and right). We have replaced the mouse images with some that we feel are more representative.

Regarding fibertype, we would predict from previous studies that the fiber type composition would be highly uniform in mouse (mostly type II as shown in Fig 4D) and mixed in man (on average approx. 50% type I:50% type II in vastus lateralis). Glycogen content in mouse muscle would also be predicted to be 1/20 of that in man. We agree with the reviewer's suggestion that fiber recruitment might explain some of the variation and attempted to perform PAS (glycogen) and phospho-p38 MAPK stainings to evaluate the uniformity of fiber recruitment. However, we are not confident in the equality of these stainings so we would prefer to keep this in mind for future testing.

For Figure 2C and 2D, it is surprising that 2DG uptake is very similar between the QUAD and soleus. Higher values would be expected both at rest and with exercise in the soleus. What accounts for the unexpected results?

We agree that the soleus usually exhibits about double the exercise-evoked 2DG uptake compared to quad in our previous work (SyLOW, Kleinert et al. 2017) (figure copy-pasted below).

We do not know the reason why they are presently more similar. One possibility could be that we presently used mice on a C57BL/10 background, not C57BL/6. Since this does not influence the conclusions made regarding NOX2, we will keep this in mind to test this in future experiments.

The in vivo glucose uptake during exercise will be greatly influenced by altered blood flow. Could the results in the KO models be related in part to altered blood flow?

We have now stained capillaries and find no genotype-difference (Fig. 4H) consistent with our previous report in Rac1 imKO (SyLOW, Kleinert et al. 2017).

Lines 235-6 refer to previous work that provided evidence that NOX2 signals independently of AMPK-TBC1D1, but reading the cited work, the evidence supporting this conclusion is not clear, and the text should more specifically identify the experimental evidence that is the basis for this statement.

Thanks for pointing out this lack of clarity. The cited work supports that NOX2 and AMPK-TBC1D1 act in parallel to regulate glucose uptake. Thus, AMPK activation does not activate Rac1 and Rac1 KO does not affect AICAR-stimulated glucose uptake. Conversely, stretch potently activates Rac1 but not AMPK and stretch-stimulated glucose uptake is not reduced in AMPK KD mice. Importantly, AICAR and stretch are additive stimuli with each-other and with insulin but not contraction. This suggests that contraction *ex vivo* is mediated by a stretch-component and an AMPK-component. We cited a few extra studies placing TBC1D1 in the AICAR-stimulated pathway to glucose uptake.

To make this clearer but still keep it brief, we have reworded:

“Worth noting, we believe NOX2 to signal independently of the AMPK-TBC1D1 signaling axis, based on the previous work by us and others 14, 31, 32, 33, 34”

To

“Worth noting, we believe Rac1 to signal in parallel to the AMPK-TBC1D1 signaling axis, based on the previous work by us and others 14, 31, 32, 33, 34. In conjunction with the current study, strongly suggesting that Rac1 regulates glucose uptake via NOX2, we propose that NOX2 also signals independently of AMPK to regulate glucose uptake during moderate intensity exercise”

Citation #31 states that Rac1 accounts for the majority of muscle glucose uptake with ex vivo contraction, but not in vivo exercise. This result appears to be in conflict with the current results. Please clarify.

We respectfully believe that you have misread/understood our previous work. In the cited paper, we cross-bred Rac1 and AMPK-deficient mice and found a partial reduction in exercise-stimulated glucose uptake in vivo in Rac1 KO mice and no effect of AMPK KD. Ex vivo, both pathways seemed involved in contraction-stimulated glucose transport. The *in vivo*-data are consistent with our previous observations in Rac1 KO mice (Sylov, Kleinert et al. 2017) and AMPK KD mice (Maarbjerg, Jorgensen et al. 2009) . Hence, the data are very much in agreement with the current study.

Lines 175-6 states that exercise-induced ROS production is suggested to activate a number of kinases linked to glucose uptake-regulation in muscle based on citations #23. However, looking at that citation, it is not obvious which specific kinases are being referred to. Please clarify.

Thanks. Wrong reference. We have now cited (He, Li et al. 2016) which states e.g.

As mentioned previously, MAPK also plays an important role in exercise-induced adaptation in skeletal muscle. MAPK is composed of four subfamilies (ERK1/2, JNK, p38 MAPK, and ERK5) (Kramer and Goodyear, 2007). The activities of ERK and MEK have a positive correlation with exercise intensity in human skeletal muscle (Widegren et al., 2000). ROS such as H₂O₂, can induce the activation of ERK, JNK, and p38 MAPK in skeletal myoblasts in a dose- and time-dependent manner (Kefaloyianni et al., 2006). Oxidative stress could also modulate the MAPK signaling pathway through insulin signaling and glucose transport (Kim et al., 2006; Sandström et al., 2006; Kramer and Goodyear, 2007).

Citation #23 claims that “exercise at moderate intensity (50–75%VO₂max), the primary source of ROS is from the mitochondria.” Please clarify how this statement corresponds to the results in this study.

We believe this statement is wrong since mitochondria have been convincingly shown not to contribute to ROS production during in vitro muscle contraction (Michaelson, Shi et al. 2010). Furthermore, we have now directly measured oxidation of roGFP-Orp1 targeted to mitochondria and see a decrease rather than an increase in mitochondrial ROS in response to moderate intensity exercise *in vivo* (Fig. 2).

For the purpose of assisting with researchers who wish to reproduce the work, some minor sources

of confusion include the description on line 263 of “injected subcutaneously in FDB.” Was the solution actually injected into the FDB itself subcutaneously near the FDB?

Into the foot near but not into the FDB muscle. We have clarified this in the methods.

On lines 370-2, there is description of freezing tissue embedded in tissue-tek. Were these samples used for western blotting?

No, these were separate TA. We have clarified this in the methods.

Line 602 – please define IQR.

Inter-quartile range. Now defined. Thanks.

Reviewer #3 (Remarks to the Author):

This is a potentially important paper describing the role of NADPH oxidase 2 in skeletal muscle glucose uptake in rodents. The data presented are generally compelling and the use of specific knockout models provides reassurance on the in vivo biological relevance of the findings.

There are however some substantial concerns with the methods used in one part of the manuscript, specifically the assessments of muscle ROS production and NADPH oxidase activation:

The authors have assessed muscle ROS production ex vivo by assessment of DCF oxidation (presumably they used the reduced form, usually described as DCFH) in cryostat sections of human muscle biopsies or TA muscles from mice.

Thanks for correcting this mistake. We have changed DCF to DCFH throughout

Muscle sections were incubated with DCFH and allowed to dry overnight at room temperature. While this method has been previously published, the assay is not validated and it is not clear what is being measured or where the species that oxidise the DCFH derive from. Thus ROS (superoxide) generation during contractile activity of muscle occurs very rapidly and this species and reactive species derived from them degrade very rapidly. Since the assay is performed on muscle ex vivo, this assay cannot be directly measuring ROS generation that has occurred in vivo. Furthermore the sections are effectively incubated with the DCFH over a period up to 24 hours which allows a great deal of time for spurious DCFH oxidation deriving from many sources including (for instance) iron delocalisation within the cryosections. It may be that the assay can provide some measure of the susceptibility of the muscle sections to oxidation, but it cannot be an assay for ROS generation in muscle during exercise.

We agree with the reviewer that the post-mortem measurement of DCFH oxidation is not straightforward to interpret and must clearly measure some consequence (s) of oxidation rather than being a direct measure of ROS. That said, we still think that measuring post-mortem DCFH oxidation in this context must be reflective of NOX2-dependent ROS production during exercise since the

increase in DCFH oxidation by exercise is completely absent in NOX2-deficient *Rac1* and *ncf1** mice. A previous study using DCFH in cryosections in e.g. muscle incubated +/- contraction +/- NAC also showed that the DCFH oxidation with contraction is completely blocked by NAC, arguing that it is ROS-dependent (Merry, Steinberg et al. 2010)

Whatever the cause and inherent artifacts with this assay may be (e.g. iron de-localization), it is hard to interpret an acute increase in DCFH oxidation with exercise in WT but not in NOX2-deficient mice as anything other than a reflection of the NOX2-dependent ROS production that occurred during the exercise. Hence, we believe that DCFH oxidation reflects total oxidants production and find it valid to state this. However, to address the point raised we have inserted a “likely” in the DCFH data interpretations throughout.

However, to further substantiate the data we have now also measured cytosolic roGFP oxidation with *in vivo* exercise. These data show that NOX2 is required for exercise-induced cytosolic ROS production, similar to what is observed with DCFH.

The assay for activation of Nox2 enzymes based on oxidation of p47-roGFP is novel and based on a recently described histological approach to examine the redox status of ro-GFP based probes. However this approach does not appear to have been applied to the p47-roGFP probe previously and hence the reliability and validity of the assay has not been established. Some validation of the assay is required. It is also apparent that there is heterogeneity of fluorescence between fibres (how is this dealt with in the quantification) and the pattern of fluorescence seen in the images is not reflected in the summary graphs.

The redox histology method is general for all roGFP probes. The NEM treatment protects against reduction of oxidized roGFP2 because NEM generally alkylates free thiols, thus preventing reduction of oxidized roGFP2 by disulfide exchange with thiols contained within the tissue section. To address the reviewer’s concern, we include now our pilot experiments testing the redox histology in p47phox transfected muscles. As shown in Fig S3, the biosensor oxidized by H₂O₂ and reduced by DTT.

Reviewer #4 (Remarks to the Author):

In this study, investigators set out to examine the role of NOX2 in generating ROS in skeletal muscle during moderate exercise, and importance for glucose metabolism.

Overall this paper is extremely well executed in terms of directness of addressing the research question, adequate experiments, intuitive and well designed figures, as well as a very good text flow and excellent lingual command.

Abstract and introduction is clear and well written. The investigation of the function of ROS during exercise in skeletal muscle is very interesting, and as the authors mention, underinvestigated previously.

Thanks for the nice comments

Methods.

The authors start by framing the research question in an investigation using muscle biopsies from healthy volunteers (n=3, in this case data are sufficiently clear to prove the point). Murine models and gene transfer. Inducible muscle-specific male Rac1 mice and littermate control mice were used, with inducible gene expression under doxycyclin control. This nicely produces both the experimental group and the controls. Electroporation was used for gene transfer which is a clever choice for this study. However use of hyaluronidase seems unnecessary (electroporation alone has been deemed sufficient to obtain high level gene transfer). The amount of DNA is very high (20 or 40 µg for the tibialis anterior muscle). 5µg (or even less) should have been sufficient.

Thanks for these suggestions. We agree that for FDB electroporation hyaluronidase might be unnecessary, but for TA muscle, our pilot experiments indicated that hyaluronidase improved transfection efficiency.

It is not clear if the authors performed empty vector controls, which should be included.

Thank you for this suggestion. We would agree if we were investigating the effect of the electroporated protein on an endogenous muscle endpoint. However, with biosensors we do not see the point of including a vector-control since the biosensor is what we measure, and this would simply give no signal.

It is not stated how much time passed from the gene electrotransfer to experiments. Statistical methods in order and appropriately described.

We have now clarified this in the methods

Figures

Over all very well laid out and easily interpretable. Microscopy clear.

Minor comments:

Fig 1. By indicating 'human' and 'murine' eg by the x-axis this important information would be legible in the figure.

Done

Fig 2a. X-axis labels not intuitive (why the .00 ?)

These are time-points during a 24h day-period. We believe the 4-digit X-axis is correct. To avoid confusion, we have now labelled the axis "Time of day"

the legend is more appropriately placed in the right hand corner - not in different colored sections of the graph.

Thanks, we moved the ncfl* out of the grey box.

Minor

Line 197 - why the comma? Insulin-stimulated, muscle contraction

This is a mistake. Should read “insulin, contraction and passive stretch-induced...” Now corrected

References

- Anderson, E. J., M. E. Lustig, K. E. Boyle, T. L. Woodlief, D. A. Kane, C. T. Lin, J. W. Price, 3rd, L. Kang, P. S. Rabinovitch, H. H. Szeto, J. A. Houmard, R. N. Cortright, D. H. Wasserman and P. D. Neufer (2009). "Mitochondrial H₂O₂ emission and cellular redox state link excess fat intake to insulin resistance in both rodents and humans." *J Clin Invest* **119**(3): 573-581.
- Fukawa, T., B. C. Yan-Jiang, J. C. Min-Wen, E. T. Jun-Hao, D. Huang, C. N. Qian, P. Ong, Z. Li, S. Chen, S. Y. Mak, W. J. Lim, H. O. Kanayama, R. E. Mohan, R. R. Wang, J. H. Lai, C. Chua, H. S. Ong, K. K. Tan, Y. S. Ho, I. B. Tan, B. T. Teh and N. Shyh-Chang (2016). "Excessive fatty acid oxidation induces muscle atrophy in cancer cachexia." *Nat Med* **22**(6): 666-671.
- He, F., J. Li, Z. Liu, C. C. Chuang, W. Yang and L. Zuo (2016). "Redox Mechanism of Reactive Oxygen Species in Exercise." *Front Physiol* **7**: 486.
- Maarbjerg, S. J., S. B. Jorgensen, A. J. Rose, J. Jeppesen, T. E. Jensen, J. T. Treebak, J. B. Birk, P. Schjerling, J. F. Wojtaszewski and E. A. Richter (2009). "Genetic impairment of AMPK α 2 signaling does not reduce muscle glucose uptake during treadmill exercise in mice." *Am J Physiol Endocrinol Metab* **297**(4): E924-934.
- Merry, T. L., G. R. Steinberg, G. S. Lynch and G. K. McConell (2010). "Skeletal muscle glucose uptake during contraction is regulated by nitric oxide and ROS independently of AMPK." *Am J Physiol Endocrinol Metab* **298**(3): E577-585.
- Michaelson, L. P., G. Shi, C. W. Ward and G. G. Rodney (2010). "Mitochondrial redox potential during contraction in single intact muscle fibers." *Muscle Nerve* **42**(4): 522-529.
- Sylov, L., M. Kleinert, E. A. Richter and T. E. Jensen (2017). "Exercise-stimulated glucose uptake - regulation and implications for glycaemic control." *Nat Rev Endocrinol* **13**(3): 133-148.

Reviewers' Comments:

Reviewer #1:

Remarks to the Author:

I have no further comments to the authors.

Reviewer #2:

Remarks to the Author:

The paper's major claims are that NOX2 is a major muscle source of ROS that regulates muscle GLUT4 translocation and glucose uptake during moderate intensity exercise, and that the role of Rac1 in regulating these processes are due to Rac1's essential role in NOX2 activation. Several interesting approaches are included to support these claims. The authors have added new data in the revised manuscript. However, several comments or questions from before were addressed only in the author's response without any change in the text to address the issues in question. The text of the manuscript needs to include revisions related to the issues below.

The text in the manuscript should directly address the obvious difference between human VL and mouse TA muscle for exercise effects on DCF fluorescence in figures 1A and 1C. In the human exercise section, there is uniform signal across all fibers. In the mouse exercise section, some fibers have much greater signal than other fibers. As the authors noted in response to this comment, the mouse muscle would be expected to have less homogenous fiber type. There should be an acknowledgement and possible explanation regarding the species differences.

The authors mention that glucose delivery helps coordinate in vivo glucose uptake. The in vivo glucose uptake during exercise will be influenced by the exercise effect on blood flow. Measuring capillary density provides some information about the capacity for glucose delivery, but the text should also acknowledge that in vivo glucose uptake during exercise will be influenced by blood flow, and the effect of genotype on blood flow during exercise was not measured.

The methods for GLUT4 translocation should provide information on the confocal microscope used for the analysis and how the signals were quantified. The Methods and Legend for Figure 5 should specify if the sample size refers to the numbers of mice studied and also to indicate how many fibers were studied per mouse for GLUT4 translocation. The statistical analysis used should be identified.

Reviewer #3:

Remarks to the Author:

The authors have satisfactorily dealt with some of the points raised in the original review.

The new experiments utilizing mito-roGFP and cyto-roGFP are excellent and the authors are to be congratulated in obtaining this novel data for this revision.

I still have significant concerns about the inclusion of Figures describing the results from DCFH oxidation over 24 hours in sections of muscle. The authors argument that this assay must tell us something since it provides the expected result is a circular argument that cannot be sustained.

Reviewer #4:

Remarks to the Author:

The comments have been addressed adequately.

REVIEWERS' COMMENTS:

Reviewer #1 (Remarks to the Author):

I have no further comments to the authors.

Reviewer #2 (Remarks to the Author):

The paper's major claims are that NOX2 is a major muscle source of ROS that regulates muscle GLUT4 translocation and glucose uptake during moderate intensity exercise, and that the role of Rac1 in regulating these processes are due to Rac1's essential role in NOX2 activation. Several interesting approaches are included to support these claims. The authors have added new data in the revised manuscript. However, several comments or questions from before were addressed only in the author's response without any change in the text to address the issues in question. The text of the manuscript needs to include revisions related to the issues below.

We apologize for our negligence. As detailed below, we expanded the Discussion section to address this, including data obtained during the review process

The text in the manuscript should directly address the obvious difference between human VL and mouse TA muscle for exercise effects on DCF fluorescence in figures 1A and 1C. In the human exercise section, there is uniform signal across all fibers. In the mouse exercise section, some fibers have much greater signal than other fibers. As the authors noted in response to this comment, the mouse muscle would be expected to have less homogenous fiber type. There should be an acknowledgement and possible explanation regarding the species differences.

We have put in an acknowledgement of the potential species-difference in DCFH staining pattern in the Discussion p. 10. We have not systematically investigated this so we currently have no explanation.

The authors mention that glucose delivery helps coordinate in vivo glucose uptake. The in vivo glucose uptake during exercise will be influenced by the exercise effect on blood flow. Measuring capillary density provides some information about the capacity for glucose delivery, but the text should also acknowledge that in vivo glucose uptake during exercise will be influenced by blood flow, and the effect of genotype on blood flow during exercise was not measured

We now discussed this in the Discussion p. 11.

The methods for GLUT4 translocation should provide information on the confocal microscope used for the analysis and how the signals were quantified. The Methods and Legend for Figure 5 should specify if the sample size refers to the numbers of mice studied and also to indicate how many fibers were studied per mouse for GLUT4 translocation. The statistical analysis used should be identified.

Thanks to the reviewer for pointing out this, now we added more information about the methodology we used to detect membrane GLUT4 in both legend and methods section. We also include some missing statistical information in the legend and the p values in the Source data file.

Reviewer #3 (Remarks to the Author):

The authors have satisfactorily dealt with some of the points raised in the original review.

The new experiments utilizing mito-roGFP and cyto-roGFP are excellent and the authors are to be congratulated in obtaining this novel data for this revision.

Thank you

I still have significant concerns about the inclusion of Figures describing the results from DCFH

oxidation over 24 hours in sections of muscle. The authors argument that this assay must tell us something since it provides the expected result is a circular argument that cannot be sustained.

Biochemical assays are usually validated via positive and negative control experiments. Here, we show that DCFH oxidation is increased by in vivo exercise and more critically that this increase is absent in mice lacking functional NOX2. This shows that the DCFH assay detects an NOX2-dependent exercise-induced increase in DCFH oxidation. This is not a circular argument since we provide empirical evidence distinct from the conclusion.

We think the human data in conjunction with the assay-validation in mice are an interesting and highly relevant translational extension and we would like to keep them in the manuscript. To address the reviewer's concern and the fact that measuring ROS in human muscle is challenging, we now discuss the limitations and concerns about the methodology in the Discussion. We hope this helps to encourage development of methods for assessing ROS in human muscle during in vivo conditions.

Just to clarify, the process is not overnight and therefore not 24 hours but around 8h.

Reviewer #4 (Remarks to the Author):

The comments have been addressed adequately.